# The therapeutic potential of phage pT2784 against ST40-KL47 type *Acinetobacter baumannii* and bacterial fitness trade-offs

Zihe Zhou,[1] Hanyu Fu,[2] Zhongyu Han,[1] Mengzhe Li,[3] Yigang Tong,[3] Ning Shen,[2] Jiajia Zheng[1]

**ABSTRACT** *Acinetobacter baumannii* usually causes high-mortality infections as the World Health Organization's priority pathogen. Herein, we isolated a novel lytic phage pT2784 from hospital wastewater, specifically targeting the emerging *Acinetobacter baumannii* ST40-KL47 lineage and demonstrated significant antibacterial effects in *in vitro* and *in vivo* models. Morphological analysis showed that pT2784 is a myovirus with a head diameter of 54.53 ± 0.27 nm and a tail length of 71.96 ± 0.87 nm. Genomic analysis revealed that pT2784 possessed a 44,335 bp DNA genome with low identity (≤61%) to known phages, supporting its classification into a new genus within *Caudoviricetes* class. Phage pT2784 displayed an optimal multiplicity of infection of 0.01, a short latent period of approximately 5 min, and a burst size of 193 plaque-forming units (PFU)/cell. It maintained high stability over a broad temperature range (4°C–65°C) and pH range (2–11). Phage-resistant strains were isolated and purified following 24-h co-culture. Whole-genome sequencing identified missense mutations in *itrA3* and *gtr50*, which encoded key enzymes involved in the early-stage pathway of capsule polysaccharide synthesis. Phenotypic analysis confirmed that phage-resistant strains exhibited capsule thinning, impaired phage adsorption, significantly reduced biofilm formation, and attenuated virulence, evidenced by over 80% survival in *Galleria mellonella* at 72 h, while none survived in the wild-type group by 12 h. Collectively, this study not only presented pT2784 as a promising therapeutic candidate but also elucidated the mechanism by which evolution under phage pressure promoted attenuated virulence in exchange for resistance, offering a crucial theoretical basis for developing phage-based interventions against *Acinetobacter baumannii*.

**IMPORTANCE** The rise of antibiotic-resistant *Acinetobacter baumannii* is a critical global health threat, urgently demanding new treatments. This study introduced a newly discovered phage, pT2784, which exhibited potent and specific lytic activity against *Acinetobacter baumannii* ST40-KL47. Crucially, we demonstrated that upon the emergence of phage resistance, the bacteria concurrently underwent a significant attenuation of their pathogenic potential. This work not only provided a promising new candidate for phage therapy but also revealed a fundamental trade-off where the path to phage resistance leads to attenuated bacterial virulence, offering a strategic advantage for managing challenging *Acinetobacter baumannii* infections.

**KEYWORDS** *Acinetobacter baumannii*, phage, phage resistance, capsule polysaccharide, fitness cost

*A*cinetobacter baumannii (*A. baumannii*) is ranked at the highest level on the World Health Organization's list of priority pathogens (1). It is a leading cause of hospital-acquired infections, particularly in intensive care units, where it is associated with high mortality in cases of pneumonia and bloodstream infections (2). Capsular polysaccharides (CPS) are one of the important virulence factors of *A. baumannii* (3), and most clinical

**Peer Reviewer** Wu Nannan, Fudan University, Shanghai, China

Address correspondence to Mengzhe Li, futurelmz123@163.com, Yigang Tong, tongyigang@mail.buct.edu.cn, Ning Shen, puh3shenning@bjmu.edu.cn, or Jiajia Zheng, zhengjiajia@bjmu.edu.cn.

Zihe Zhou and Hanyu Fu contributed equally to this article. Author order was mutually agreed upon by the co-first authors.

The authors declare no conflict of interest.

See the funding table on p. 15.

isolates produce a thick surface polysaccharide capsule (4). This structure confers broad protection against antibiotics, disinfectants, and environmental pressures (5), enabling long-term persistence on medical surfaces and thereby increasing the risk of hospital-acquired infections (6).

The remarkable genetic plasticity of *A. baumannii* has resulted in extensive strain diversity among clinical isolates (7–11). Epidemiological surveillance in China has identified ST2_Pasteur through multi-locus sequence typing (MLST) as the predominant clone responsible for hospital-acquired *A. baumannii* infections. ST40_Pasteur and ST164_Pasteur have emerged as an increasingly prevalent clone, with its clinical isolation rate continuously increasing in the past 3 years (7, 12–14). The gene encoding CPS on the K locus is strain-specific, allowing for a huge diversity in the composition of *A. baumannii* capsules. Current reports have documented at least 120 distinct K locus variants and approximately 40 structurally unique polysaccharides (8, 15). The KL47 capsular type is widely distributed not only in the predominant ST2 lineage but also in other clinically prevalent lineages, including ST40 and ST164. Critically, KL47 *A. baumannii* isolates within these lineages frequently demonstrate a multidrug-resistant phenotype (16, 17), posing a significant threat to healthcare systems. *A. baumannii* phages primarily utilized the CPS as their receptors (18, 19). Although over 10 CPS-targeting phages have been characterized, such as phage Highwayman targeting ST2-KL3 strains, phages Margaret and Fishpie infecting ST2-KL2 strains, and phage Tama, which could recognize ST2-KL9 capsule type (18), phage resources specifically targeting the KL47 strains remain remarkably scarce.

Accumulating evidence demonstrates that phages, whether administered as monotherapy or in combination with antibiotics, can effectively combat *A. baumannii* infections (20–24). Despite its considerable therapeutic promise, phage therapy remains constrained by substantial challenges that impede its broad clinical adoption. The expeditious development of phage resistance poses a central obstacle, stemming from either the structural alteration of surface receptors or the deployment of intracellular defense systems, including restriction modification (R-M) and abortive infection (Abi) systems (25, 26). Mutations that alter or eliminate phage receptors constitute the primary mechanism of phage resistance in *A. baumannii* (27–29).

In this study, we have isolated a novel *A. baumannii* phage and conducted a comprehensive analysis of its genomic sequence, biological characteristics, and therapeutic potential *in vitro* and *in vivo*. The mechanism and fitness cost of phage resistance were further explored. Our work can extend beyond providing novel phage resources against *A. baumannii* infections for clinical therapy to dialectically understand the pros and cons of phage resistance.

## RESULTS

### Phage isolation and its morphology

Using clinically isolated *A. baumannii* T2784 as the host, a novel phage, designated as pT2784, was isolated from hospital sewage samples. After three rounds of purification, it formed transparent circular plaques with uniform size of 0.96 ± 0.06 mm on LB soft agar, surrounded by turbid inhibition zones (Fig. 1A). Transmission electron microscopy (TEM) showed that phage pT2784 had a head with a diameter of 54.53 ± 0.27 nm and a tail measuring 71.96 ± 0.87 nm in length (Fig. 1B).

### Host range

The host range analysis of phage pT2784 was performed against a diverse panel of 36 *A. baumannii* clinical isolates spanning 19 distinct ST-KL types, which included both multidrug-resistant (MDR)/extensively drug-resistant (XDR) and susceptible strains. Notably, the phage exhibited strict specificity for ST40-KL47 type *A. baumannii* strains, while it showed no infectivity toward the other genotypes and other common nosocomial pathogens, such as *Klebsiella pneumoniae* (*K. pneumoniae*) and *Pseudomonas aeruginosa* (*P. aeruginosa*), as detailed in Table 1. The host strain T2784 was isolated from

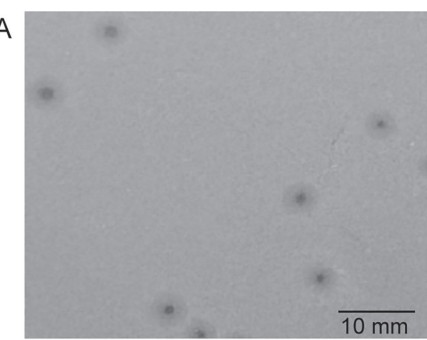
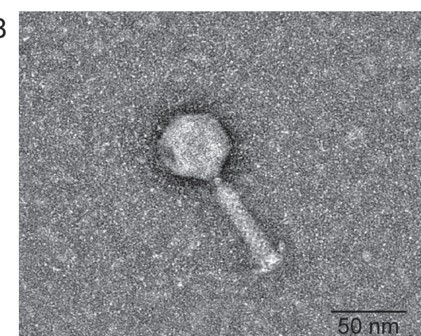

**FIG 1** Morphology of phage pT2784. (A) Plaque morphology of phage pT2784 on LB soft agar. (B) TEM image of phage pT2784.

a patient's respiratory sample and was relatively susceptible to commonly used antibiotics. Other strains of the same ST40-KL47 lineage have also been recovered from the patient's bed unit environment. Antibiotic susceptibility profiles of all strains used in the host range test are provided in Table S1.

## Genome analysis of phage pT2784

Phage pT2784 had a DNA genome with a length of 44,335 bp and a GC content of 37.82% (Fig. 2A). Genomic annotation predicted 76 putative open reading frames (ORFs), among which 43 encoded functional proteins (Table 2). These ORFs were categorized into five essential processes: phage structural composition, DNA replication and repair, transcription and translation, phage packaging, and host lysis. These included 21 structural proteins (ORF 17, ORF27-29, ORF31-32, ORF34, ORF36-40, ORF43-46, ORF48-52), nine proteins associated with transcription and translation (ORF4, ORF6, ORF11, ORF33, ORF56, ORF61, ORF64-65, ORF74), three packaging proteins (ORF14-16), six proteins related to DNA replication and repair (ORF2, ORF35, ORF60, ORF68, ORF70-71), and four host lysis proteins (ORF41, ORF42, ORF53-54). The remaining ORFs encoded hypothetical proteins with unknown functions (Table S2). No lysogenic-related genes were found in the genome of pT2784, representing it as a potential candidate with clinical application values.

To verify the novelty of phage pT2784, a basic local alignment search tool (BLASTn) search was performed, and the average nucleotide identity (ANI) was calculated against the other 11 *A. baumannii* phages sharing similarity with pT2784 (Fig. 2B). The highest genomic similarity was 61% with phage vB_AbaM_IME284 (MH853787.1), which is below the 70% genus delineation threshold set by the International Committee on Taxonomy of Viruses (30, 31), supporting its placement within a novel genus under *Caudoviricetes*; *Caudoviricetes incertae sedis*. Whole-genome phylogeny (Fig. 2C) positioned pT2784 within a monophyletic clade alongside *A. baumannii* phage AP22 (NC_017984.1). While phylogenetic analysis based on the terminase large subunit (Fig. 2D), which included the 13 most similar sequences, placed pT2784 within the same clade as *Obolenskvirus* phage Abp95 (MZ618622.1). Despite this conserved functional gene association, the low whole-genome ANI supported the classification of pT2784 as a novel genus.

## Biological characterization of phage pT2784

The multiplicity of infection (MOI) is defined as the ratio of plaque-forming units (PFU) of phages to colony-forming units (CFU) of bacterial cells at the time of infection initiation (32). Phage pT2784 exhibited robust propagation across a broad MOI range (Fig. 3A). The highest phage titer ($1.07 \times 10^9$ PFU/mL) was observed at an MOI of 0.01, which was consequently identified as the optimal MOI.

The one-step growth curve (Fig. 3B) showed that the latent period of phage pT2784, from phage adsorption to bacterial surface to the release of the first batch of progeny

**TABLE 1** Host range of pT2784*a*

| Species | Strain | ST type | KL type | MDR/XDR status | Phage sensitivity*b* |
|---|---|---|---|---|---|
| *A. baumannii* | T12965 | 2 | 2 | XDR | − |
| *A. baumannii* | T12967 | 2 | 2 | XDR | − |
| *A. baumannii* | T13520 | 2 | 2 | XDR | − |
| *A. baumannii* | T3048 | 2 | 3 | XDR | − |
| *A. baumannii* | T3094 | 2 | 3 | XDR | − |
| *A. baumannii* | T3307 | 2 | 3 | XDR | − |
| *A. baumannii* | T1319 | 2 | 7 | XDR | − |
| *A. baumannii* | T1658 | 2 | 7 | XDR | − |
| *A. baumannii* | T2010 | 2 | 7 | XDR | − |
| *A. baumannii* | T12430 | 2 | 9 | XDR | − |
| *A. baumannii* | T12457 | 2 | 9 | XDR | − |
| *A. baumannii* | T13166 | 2 | 9 | XDR | − |
| *A. baumannii* | T2297 | 40 | 45 | XDR | − |
| *A. baumannii* | T2784 | 40 | 47 | − | + |
| *A. baumannii* | T3153 | 40 | 47 | − | + |
| *A. baumannii* | T3311 | 113 | 139 | MDR | − |
| *A. baumannii* | T3840 | 113 | 139 | MDR | − |
| *A. baumannii* | T4258 | 113 | 139 | MDR | − |
| *A. baumannii* | T10332 | 571 | 10 | XDR | − |
| *A. baumannii* | T10408 | 571 | 10 | XDR | − |
| *A. baumannii* | T11233 | 2 | 104 | XDR | − |
| *A. baumannii* | T1058 | 2 | 104 | XDR | − |
| *A. baumannii* | T2977 | 2 | 104 | XDR | − |
| *A. baumannii* | T10252 | 2 | 101 | XDR | − |
| *A. baumannii* | T10331 | 2 | 101 | XDR | − |
| *A. baumannii* | T11889 | 2 | 101 | XDR | − |
| *A. baumannii* | T13697 | 2 | 160 | XDR | − |
| *A. baumannii* | T1005 | 2 | 160 | MDR | − |
| *A. baumannii* | T11280 | 221 | 14 | MDR | − |
| *A. baumannii* | T10858 | 284 | 14 | − | − |
| *A. baumannii* | T11572 | 396 | 38 | MDR | − |
| *A. baumannii* | T10236 | 464 | 210 | XDR | |
| *A. baumannii* | T10780 | 119 | 100 | MDR | − |
| *A. baumannii* | T11867 | 57 | 230 | − | − |
| *A. baumannii* | T10223 | 354 | 125 | − | − |
| *A. baumannii* | F7320 | 23 | 108 | − | − |
| *K. pneumoniae* | 217370 | 11 | 47 | XDR | − |
| *K. pneumoniae* | T5003 | 11 | 64 | XDR | − |
| *P. aeruginosa* | T3475 | 162 | − | MDR | − |
| *P. aeruginosa* | T3553 | 1196 | − | − | − |

*a*MLST-Pasteur typing was used. MDR, multidrug-resistant; XDR, extensively drug-resistant.
*b*"+" indicates that the isolate was susceptible to the phage; "−" indicates that the isolate was not XDR/MDR or was not susceptible to the phage.

phages, was about 5 min. Then, it entered the burst phase, which lasted for about 35 min. The burst size of phage was calculated as 193 PFU/cell by dividing the difference between the final and initial concentrations of phages by the initial concentration of bacteria. After 40 min, it reached the plateau period.

The titer of phage pT2784 remained at least $1 \times 10^8$ PFU/mL at temperatures ranging from 4°C to 65°C, while the titer decreased significantly at 75°C and declined to 0 at 85°C (Fig. 3C). Similarly, phage pT2784 was highly stable under pH 2–11 conditions and rapidly inactivates at pH 1 and 12 (Fig. 3D).

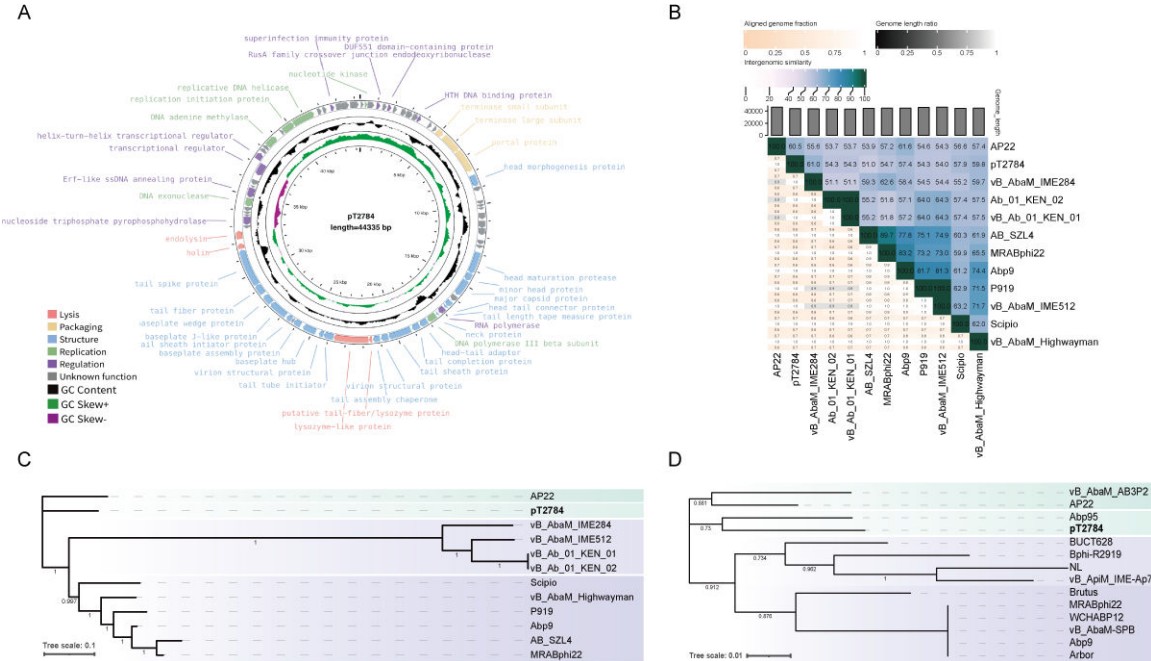

**FIG 2** Phage genome analysis. (A) Genomic annotation circle diagram of phage pT2784. (B) Genomic similarity between phage pT2784 and 11 other phages. (C) Phylogenetic tree based on the whole genome of phage pT2784. (D) Phylogenetic tree based on the terminase large subunit of phage pT2784.

## *In vitro* and *in vivo* efficacy of phages against *A. baumannii*

Phage pT2784 exhibited strong lytic activity against host bacteria *in vitro* (Fig. 4A), completely suppressing bacterial growth at first across a range of MOIs. Notably, after 6 h of incubation, an increase in optical density at 600 nm ($OD_{600}$) was observed, but much more slowly than in the bacteria-only group. This phenomenon suggested phage-resistant bacterial variants could emerge after long-term co-incubation.

In a *G. mellonella* infection model, inoculation with $5 \times 10^7$ CFU/mL *A. baumannii* resulted in progressive larval mortality, with a final survival rate of only 10% at 72 h post-infection (Fig. 4B). This dose provided a suitable observation window and was therefore selected for subsequent *in vivo* efficacy experiments.

Treatment with phage pT2784 significantly enhanced the survival of infected *G. mellonella* larvae in a dose-dependent manner, with the greatest protection observed at a high MOI of 100 (Fig. 4C). Notably, larvae receiving phage alone exhibited 100% survival, confirming the *in vivo* safety of pT2784 and underscoring its potential as an alternative antimicrobial agent.

## Impairment of early-stage pathway of capsule synthesis confers resistance to phages

To investigate the resistance mechanisms, three phage-resistant isolates, designated pT2784-R1, R2, and R3, were obtained from the 24-h co-culture. All three isolates exhibited resistance, as shown by the absence of plaques (Fig. 5A) and their growth on soft agar containing phage pT2784 (Fig. 5B).

As receptor modification or loss constitutes the primary molecular mechanism resulting in impaired phage adsorption (27, 33), we first quantitatively compared the adsorption efficiency of phage pT2784 between wild-type and phage-resistant strains. In contrast to the near-complete adsorption observed in the wild-type host, the adsorption efficiencies in the phage-resistant isolates pT2784-R1, pT2784-R2, and pT2784-R3 dropped sharply to 19.15%, 20.84%, and 24.01%, respectively (Fig. 5C through E). This marked reduction is consistent with receptor loss or structural modification as the underlying resistance mechanism.

**TABLE 2** Putative functional ORFs in phage pT2784 genome

| ORF | Strand[a] | Start | Stop | Nucleotide length (bp) | Amino acid length (aa) | Putative function |
|-----|-----------|-------|------|------------------------|------------------------|-------------------|
| ORF2 | + | 278 | 433 | 156 | 52 | Nucleotide kinase |
| ORF4 | + | 944 | 1,165 | 222 | 74 | RusA family crossover junction endodeoxyribonuclease |
| ORF6 | + | 1,608 | 1,823 | 216 | 72 | DUF551 domain-containing protein |
| ORF11 | + | 3,462 | 3,662 | 201 | 67 | HTH DNA binding protein |
| ORF14 | + | 4,810 | 5,241 | 432 | 144 | Terminase small subunit |
| ORF15 | + | 5,342 | 6,523 | 1,182 | 394 | Terminase large subunit |
| ORF16 | + | 6,527 | 7,954 | 1,428 | 476 | Portal protein |
| ORF17 | + | 8,188 | 8,895 | 708 | 236 | Head morphogenesis protein |
| ORF27 | + | 12,902 | 14,236 | 1,335 | 445 | Head maturation protease |
| ORF28 | + | 14,244 | 14,723 | 480 | 160 | Minor head protein |
| ORF29 | + | 14,733 | 15,752 | 1,020 | 340 | Major capsid protein |
| ORF31 | + | 16,251 | 16,622 | 372 | 124 | Head tail connector protein |
| ORF32 | + | 16,638 | 16,784 | 147 | 49 | Tail length tape measure protein |
| ORF33 | + | 16,829 | 17,221 | 393 | 131 | RNA polymerase |
| ORF34 | + | 17,308 | 17,403 | 96 | 32 | Neck protein |
| ORF35 | + | 17,477 | 18,034 | 558 | 186 | DNA polymerase III beta subunit |
| ORF36 | + | 18,072 | 18,575 | 504 | 168 | Head adaptor |
| ORF37 | + | 18,602 | 19,066 | 465 | 155 | Tail completion protein |
| ORF38 | + | 19,149 | 20,519 | 1,371 | 457 | Tail sheath protein |
| ORF39 | + | 20,532 | 20,981 | 450 | 150 | Virion structural protein |
| ORF40 | + | 21,027 | 21,452 | 426 | 142 | Tail assembly chaperone |
| ORF41 | + | 21,482 | 21,694 | 213 | 71 | Putative tail-fiber/lysozyme protein |
| ORF42 | + | 21,697 | 23,727 | 2,031 | 677 | lysozyme-like protein |
| ORF43 | + | 23,735 | 24,331 | 597 | 199 | Tail tube initiator |
| ORF44 | + | 24,333 | 24,611 | 279 | 93 | Virion structural protein |
| ORF45 | + | 24,798 | 25,610 | 813 | 271 | Baseplate hub |
| ORF46 | + | 25,765 | 26,238 | 474 | 158 | Baseplate assembly protein |
| ORF48 | + | 26,393 | 26,737 | 345 | 115 | Tail sheath intiator protein |
| ORF49 | + | 26,812 | 27,918 | 1,107 | 369 | Baseplate J-like protein |
| ORF50 | + | 27,942 | 28,544 | 603 | 201 | Baseplate wedge protein |
| ORF51 | + | 28,666 | 29,325 | 660 | 220 | Tail fiber protein |
| ORF52 | + | 29,327 | 31,603 | 2,277 | 759 | Tail spike protein |
| ORF53 | + | 31,659 | 31,994 | 336 | 112 | Holin |
| ORF54 | + | 32,240 | 32,752 | 513 | 171 | Endolysin |
| ORF56 | − | 33,607 | 33,074 | 534 | 178 | Nucleoside triphosphate pyrophosphohydrolase |
| ORF60 | − | 34,824 | 34,258 | 567 | 189 | DNA exonuclease |
| ORF61 | − | 35,492 | 34,857 | 636 | 212 | Erf-like ssDNA annealing protein |
| ORF64 | − | 37,169 | 36,381 | 789 | 263 | Transcriptional regulator |
| ORF65 | + | 37,244 | 37,669 | 426 | 142 | Helix-turn-helix transcriptional regulator |
| ORF68 | + | 38,256 | 38,945 | 690 | 230 | DNA adenine methylase |
| ORF70 | + | 39,525 | 40,256 | 732 | 244 | Replication initiation protein |
| ORF71 | + | 40,262 | 41,605 | 1,344 | 448 | Replicative DNA helicase |
| ORF74 | + | 42,601 | 42,813 | 213 | 71 | Immunity to superinfection |

[a]"+" indicates the forward strand; "−" indicates the reverse strand.

Comparative genomic analysis identified single-nucleotide mutations in the *itrA3* gene of pT2784-R1 (Leu to Pro at position 151) and pT2784-R3 (Gly to Arg at position 79), as well as in the *gtr50* gene of pT2784-R2 (Val to Glu at position 225) (Fig. 5F). The *itrA3* and *gtr50* genes were located within the K locus. Specifically, *itrA3* encoded the initial glycosyltransferase responsible for oligosaccharide unit initiation (34), while *gtr50* encoded a glycosyltransferase (35). These mutations likely affected the function of the encoded enzymes, impairing the early stage of capsule synthesis and conferring phage

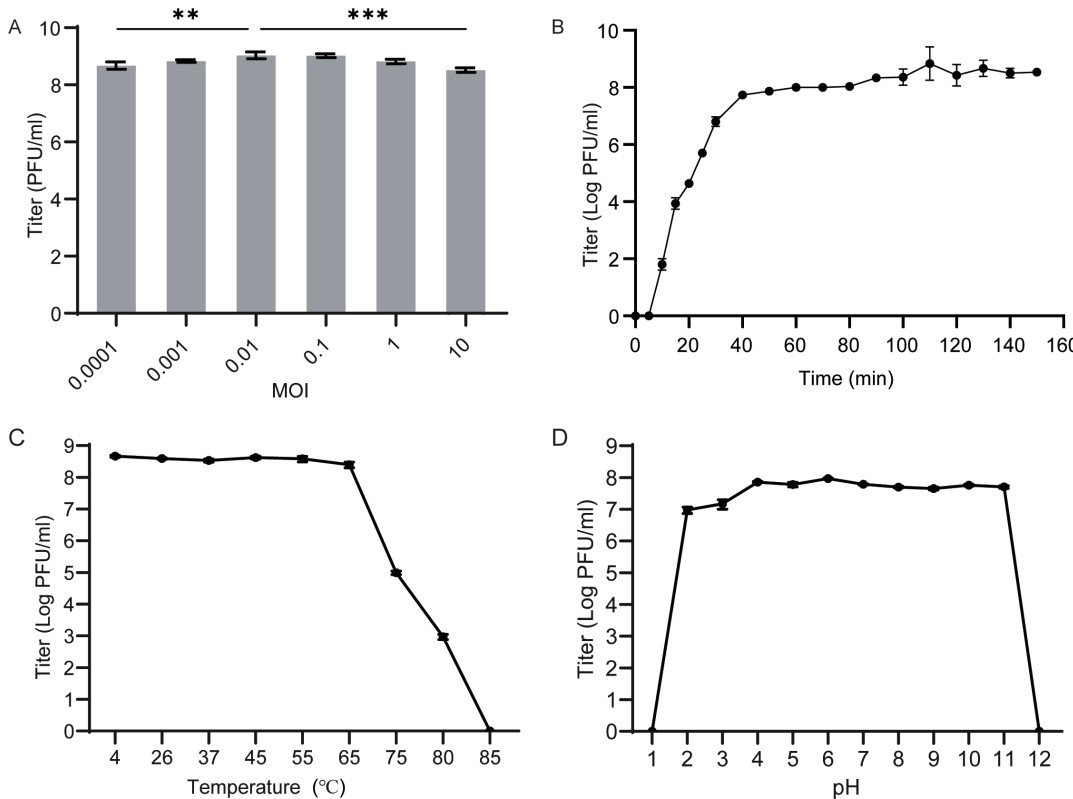

**FIG 3** Biological characterization of phage pT2784. (A) Optimal MOI assays of phage pT2784. (B) One-step growth curve of phage pT2784. (C) Temperature stability of phage pT2784. (D) pH stability of phage pT2784. **$P < 0.01$; ***$P < 0.001$.

resistance. This genetic evidence strongly supports the capsular polysaccharide as the primary receptor for phage pT2784 adsorption and subsequent infection.

Genetic complementation confirmed that *itrA3* and *gtr50* mutations are responsible for phage resistance. pT2784-R1 and pT2784-R3 with the introduction of the *itrA3* gene and pT2784-R2 complemented with *gtr50* restored phage adsorption efficiencies ($P > 0.05$, Fig. 5C through E).

## Attenuation of biofilm formation and virulence in phage-resistant isolates

Given the critical role of capsular polysaccharide as a major surface structure in *A. baumannii*, we next assessed whether its structural alterations imposed fitness trade-offs on phage-resistant strains (36–38). The phage-resistant isolates and the wild-type strain formed similar circular colonies with gray-white, smooth, and neat edges on blood agar, with no apparent morphological differences (Fig. 6A). Growth rates of phage-resistant isolates also showed no significant difference compared to that of the wild-type strain (Fig. 6B).

Scanning electron microscopy (SEM) revealed that all phage-resistant isolates exhibited increased surface rugosity compared to the wild-type strain (Fig. 6C). Furthermore, biofilm formation by the resistant isolates was significantly reduced relative to the wild type after 48 h of cultivation (Fig. 6D).

In a *G. mellonella* infection model (Fig. 6E), larvae inoculated with the wild-type strain at a concentration of $1 \times 10^8$ CFU/mL exhibited 100% mortality within 12 h. In contrast, infection with phage-resistant strains resulted in less than 20% mortality at 12 h, with final mortality not exceeding 50% after 72 h, indicating a significant attenuation of virulence in the resistant variants.

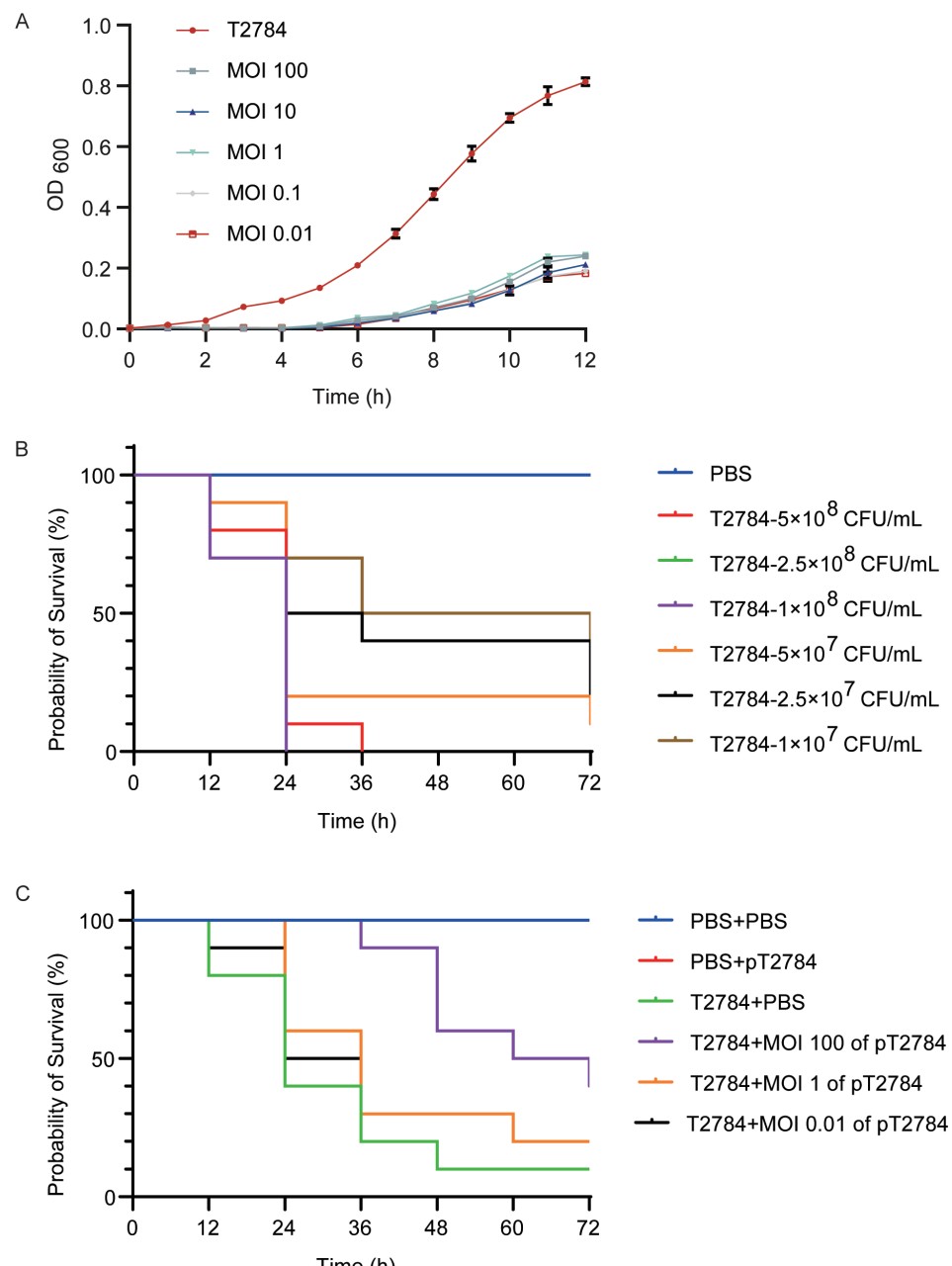

**FIG 4** Antibacterial effects of phage pT2784 *in vitro* and *in vivo*. (A) *In vitro* lysis kinetics assays of phage pT2784 at different MOI values. (B) The survival curve of *G. mellonella* infected with different inocula of *A. baumannii*. (C) *In vivo* therapeutic effects of phage pT2784.

## DISCUSSION

In this study, we isolated phage pT2784 from hospital wastewater using a clinical *A. baumannii* strain T2784 as the host. It exhibited a highly specific lytic activity, targeting only ST40-KL47 type while showing no effect against others (Table 1), which aligns with previous studies (18). Comparative analysis revealed less than 61% nucleotide identity to known phages (Fig. 2B), establishing pT2784 as a new genus under *Caudoviricetes*; *Caudoviricetes incertae sedis* (30). Notably, pT2784 exhibited a remarkably short latency period of 5 min—significantly shorter than other *A. baumannii* phages—while maintaining comparable burst sizes (39) (Fig. 3B). Further characterization demonstrated

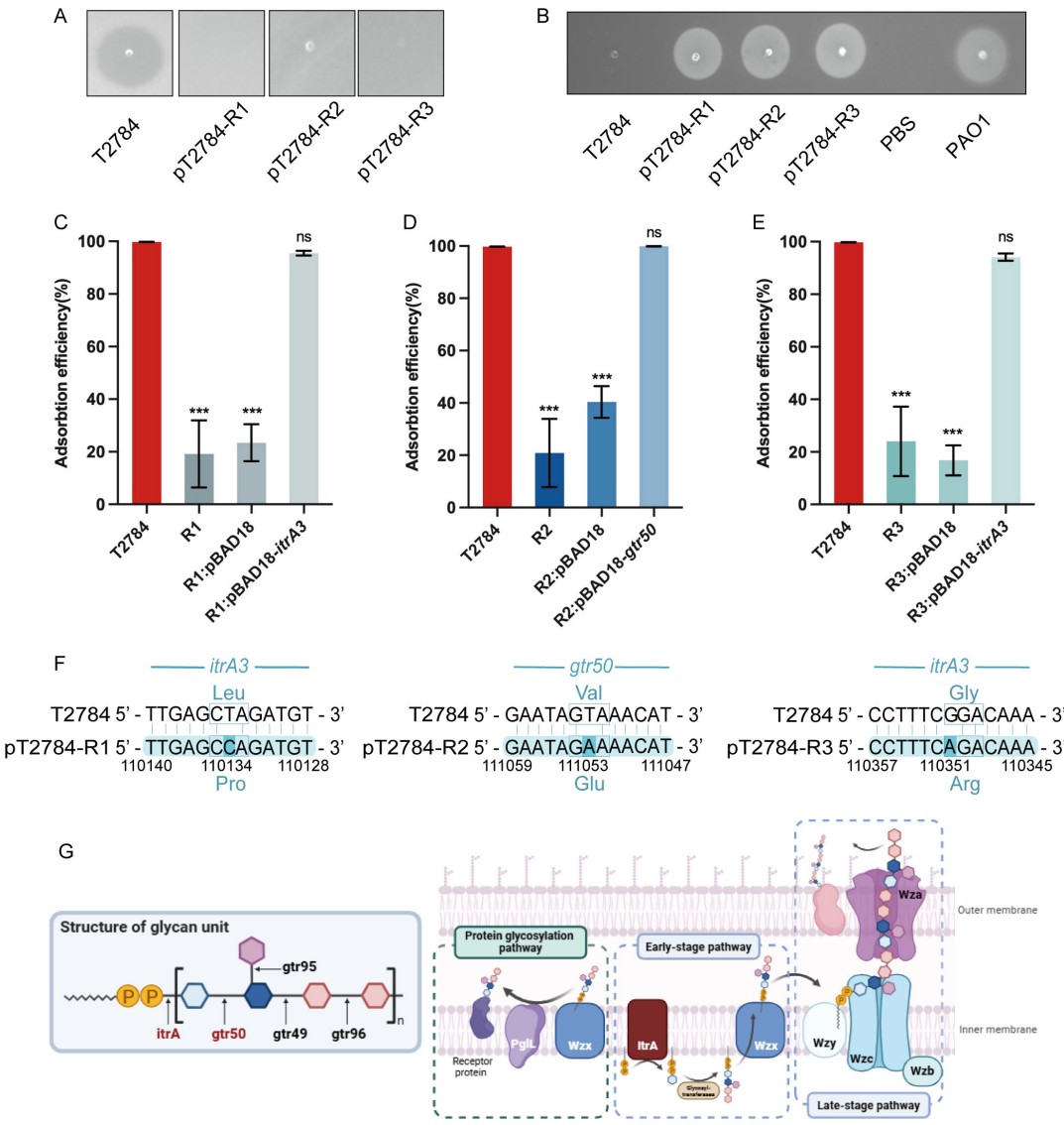

**FIG 5** Isolation, characterization, and mechanism of phage-resistant strains. (A) Spot test using phages to infect host bacteria and phage-resistant isolates. (B) Inverted spot plate assay. Bacterial suspensions of the host strain T2784 and phage-resistant isolates were spotted onto soft agar overlays containing 100 µL of phage ($10^8$ PFU/mL), while PBS was used as the negative control and *Pseudomonas aeruginosa* PAO1 was used as the positive control. (C) Adsorption efficiency of phages on host bacteria, phage-resistant isolate pT2784-R1, and complemented strain R1:pBAD-*itrA3*. (D) Adsorption efficiency of phages on host bacteria, phage-resistant isolate pT2784-R2, and complemented strain R2:pBAD-*gtr50*. (E) Adsorption efficiency of phages on host bacteria, phage-resistant isolate pT2784-R3, and complemented strain R3:pBAD-*itrA3*. (F) Detailed presentation of mutant genes. (G) Synthesis of capsule polysaccharide in *A. baumannii* consists of early and late-stage pathways. The protein O-glycosylation pathway related to biofilm formation and maturation was also depicted here. The black wavy lines represented lipid carriers, the yellow circles represented phosphates, and the repeated colored hexagons represented common repeating unit sugars. ***$P < 0.001$; ns, non-significant.

exceptional environmental resilience, including retention of infectivity at 75°C (Fig. 3C) and sustained stability across a broad pH range of 2–11 (Fig. 3D). These attributes positioned pT2784 as a promising candidate for both therapeutic applications and environmental decontamination.

The *in vivo* experiment in the *G. mellonella* model also demonstrated excellent therapeutic effects of phage pT2784 on *A. baumannii* infection, prolonging the survival time of *G. mellonella* larvae without showing any other toxic effects (Fig. 4C). Therefore, phage pT2784 had certain clinical application potential as a safe and effective therapeutic agent. Nevertheless, in the *in vitro* lysis assay with phage pT2784, resurgence of

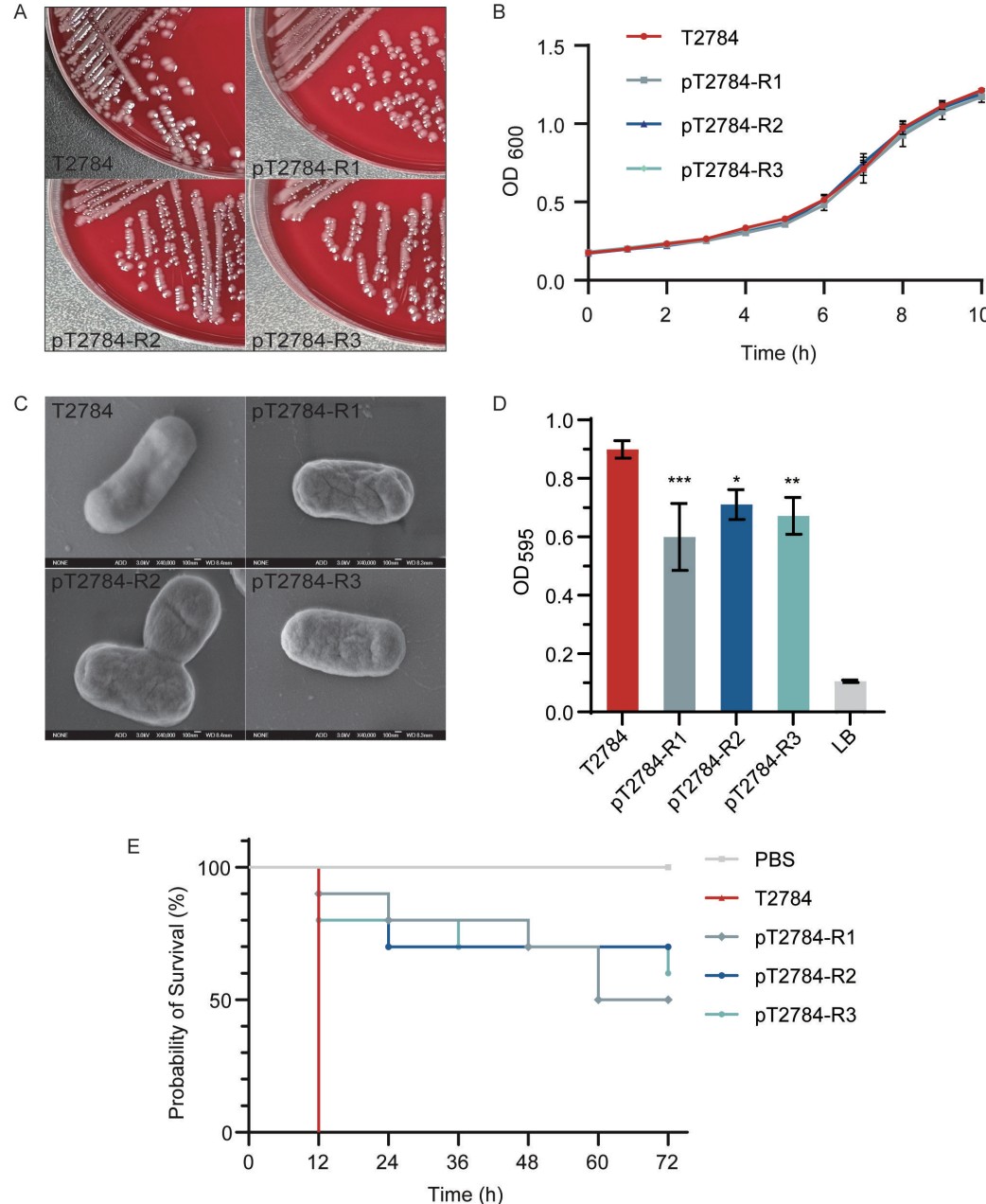

**FIG 6** Fitness cost of phage-resistant isolates. (A) The morphology of bacterial colonies on Columbia blood agar. (B) Bacterial growth curve. (C) Bacteria scanning electron microscope. (D) Bacterial biofilm formation ability. (E) The virulence of host and phage-resistant isolates. *$P < 0.05$; **$P < 0.01$; ***$P < 0.001$.

bacterial growth was observed at 6 h post-infection (Fig. 4A), coinciding with the emergence of phage-resistant isolates. Notably, Schooley et al. documented analogous resistance accompanied by marked colony morphology heterogeneity suggestive of capsule modulation during cocktail therapy for MDR *A. baumannii* infection (22). According to the "phage steering" hypothesis, the acquisition of phage resistance is often accompanied by a reduction in virulence (40). Consistent with this concept, three phage-resistant isolates were obtained and characterized in this study. These isolates exhibited markedly attenuated pathogenicity *in vivo* (Fig. 6E), supporting the trade-off between phage resistance and virulence.

Genomic analysis revealed single-nucleotide mutations in *itrA3* and *gtr50*, both encoding enzymes essential for the early-stage pathway of capsule synthesis. The *itrA3*

gene encoded an initial glycosyltransferase that primes the Und-P-linked oligosaccharide precursor, whereas *gtr50* encoded a capsule-specific glycosyltransferase responsible for extending sugar chains. Disruption of these genes likely impairs capsule assembly and alters surface receptor structures, thereby reducing phage adsorption. This effect was nearly fully reversed upon genetic complementation (Fig. 5C through E). To facilitate mechanistic interpretation, we constructed a schematic diagram (Fig. 5G) summarizing the capsule synthesis pathway in *A. baumannii*. The process could be broadly divided into early- and late-stage pathways. In the early stage, sequential glycosyltransferases assembled oligosaccharide repeating units on the cytoplasmic side of the inner membrane. In the late stage, polymerization and translocation of capsule polysaccharides to the bacterial surface were mediated by Wzy-dependent machinery. The identified *itrA3* and *gtr50* mutations occurred in the early-stage synthesis step, consistent with reduced capsule polysaccharide production observed in phage-resistant isolates. Similar loss-of-function mutations in capsule-related genes, including *gtr29*, *gpi*, and *gtr9*, have previously been associated with phage resistance in *A. baumannii* (27–29).

Impairment of capsule formation not only limited phage adsorption but also attenuated virulence (Fig. 6E). Previous studies have shown that interference with capsule biosynthesis weakens *A. baumannii* pathogenicity (41). Moreover, reduced synthesis of oligosaccharide units likely diminished O-glycosylation of surface proteins mediated by PglL, thereby impairing biofilm formation (42). Consistently, phage-resistant isolates displayed a pronounced reduction in biofilm-forming ability (Fig. 6D). Collectively, these results suggested that phage infection not only directly lysed bacteria but also exerted selective pressure driving the evolution of less virulent phenotypes through capsule-related gene mutations.

While this study provides valuable insights into the biology and therapeutic potential of phage pT2784, certain limitations should be noted. Although the host strain T2784 obtained in this study is relatively susceptible to commonly used antibiotics, it is noteworthy that both the ST40 and KL47 lineages have been widely associated with multidrug-resistant phenotypes. The convergence of these two high-risk lineages in a single strain suggests a potential risk for the evolution of resistance (Table S3). Consequently, isolating and characterizing phages targeting such strains is necessary for future therapeutic and preventive strategies. Notably, phages specific to the ST40-KL47 type of *A. baumannii* have rarely been reported, thus expanding the phage resource library against this lineage is of considerable importance. Nevertheless, we selected the strain as a representative model to explore the multifaceted interactions between phages and their hosts, including therapeutic efficacy, resistance evolution, and associated phenotypic trade-offs. Despite its antibiotic susceptibility, *A. baumannii* remains a major concern for infection control due to its remarkable environmental persistence and high transmission potential in hospital settings. Therefore, expanding phage libraries and deepening the understanding of resistance mechanisms remain essential for future translational applications.

In summary, this study identified a novel *A. baumannii* phage, pT2784, elucidated its therapeutic efficacy, characterized the genetic basis of phage resistance, and revealed a collateral trade-off between resistance and virulence. These findings provide both theoretical foundations and practical insights for the development of precision phage therapy and infection control strategies against *A. baumannii*.

## MATERIALS AND METHODS

### Bacterial strains and incubation conditions

All bacterial strains used in this study were obtained from the Department of Laboratory Medicine, Peking University Third Hospital, and stored with 25% glycerol in −80℃. For culturing, bacteria were incubated in Luria-Bertani (LB) broth overnight at 37℃ with shaking at 200 rpm. Antimicrobial susceptibility testing was performed by Vitek 2, and the results were interpreted using the 2025 Clinical and Laboratory Standards

Institute guideline breakpoints. MDR was defined as resistance to three or more different antimicrobial classes, and XDR was defined as susceptible to only one or two classes (43).

## Phage isolation and purification

Phages were isolated from sewage samples collected from Peking University Third Hospital. Sewage samples were centrifuged at 8,000 rpm for 10 min and filtered through a 0.22 µm filter. The filtrate was mixed with bacteria in LB broth and incubated at 37°C for 6 h with shaking. Then, the mixture was filtered to discard bacterial debris, and the filtrate was mixed with bacteria in LB soft agar and poured onto LB agar for incubation overnight at 37°C. Single plaques were picked up and suspended in PBS for proliferation and purification until the morphology of the plaques was consistent.

## Optimal multiplicity of infection

Phages and host bacteria were mixed at an MOI of 0.0001, 0.001, 0.01, 0.1, 1, and 10, respectively. And the mixture was added to LB broth and incubated for 6 h at 37°C with shaking. Then the mixture was filtered through a 0.22 µm filter, and a 10-fold dilution was performed. Phage titer was determined using the double-layer plate method, and the group with the highest titer is the optimal MOI.

## One-step growth curve

Phages and host bacteria were mixed at the optimal MOI, incubated at 37°C for 15 min, and then centrifuged at $12,000 \times g$ for 5 min. The precipitation was washed three times with PBS and resuspended in 50 mL LB broth. Then, 1 mL of the sample was taken immediately, which was the "0" sample. Subsequent samples were obtained at 5, 10, 15, 20, 25, 30, 40, 50, 60, 70, 80, 90, 100, 110, 120, 130, 140, and 150 min, respectively, and phage titers were determined through the double-layer plate method.

## pH and temperature stability

A buffer solution with a pH ranging from 1 to 12 was prepared using 1 M HCl and 1 M NaOH. Phages were mixed with buffer solution at different pH values and incubated at 37°C for 1 h. Similarly, phages were incubated at 4°C, 26°C, 37°C, 50°C, 65°C, 75°C, 80°C, and 85°C for 1 h, respectively. Phage titers were determined using the double-layer plate method under different pH values and temperature conditions. Each experiment was repeated three times.

## Transmission electron microscopy and scanning electron microscopy

The morphology of phages was observed under a TEM. Briefly, phages were dropped onto a carbon-coated copper mesh and were fully settled on the surface of the mesh. Phage particles were negatively stained with 2% uranium acetate and observed under a JEM-1400 (JEOL Ltd., Tokyo, Japan) TEM at 80 kV.

SEM was performed as previously described (44) with slight modifications. In short, the pellets from 5 mL of bacteria were collected, completely immersed in 2.5% glutaraldehyde fixative, and incubated overnight at 4°C. Then the bacteria were stained with 1.0% osmium tetroxide and washed thoroughly. Bacteria were dehydrated in a series of gradient ethanol solutions, followed by vacuum drying, and subsequently observed with a JSM-6700F (JEOL Ltd., Tokyo, Japan) SEM.

## Isolation and purification of phage-resistant strains

Phages were mixed with the bacteria at the optimal MOI in LB broth and incubated at 37°C for 24 h with shaking. The mixture was incubated on LB agar to obtain single colonies, and three rounds of purification were performed. Phage resistance was initially demonstrated using the double-layer plate method. Suspected phage-resistant strains (200 µL) were added to LB soft agar and poured onto LB agar. A volume of 2 µL of phage

was spotted onto the bacterial lawn on soft agar. Bacteria showing no clear plaques were considered phage-resistant strains. In addition, we also used the reverse spot method described by Gordillo Altamirano et al. (27) to further confirm phage resistance. Host bacteria, suspected phage-resistant strains, *Pseudomonas aeruginosa* PAO1, and PBS (2 µL) were spotted onto soft agar containing phages at a titer of $10^8$ PFU/mL, then incubated overnight at 37°C. Bacteria that survived on soft agar with phages were confirmed as phage-resistant strains.

## Phage adsorption assay

Phage was mixed with bacteria at an MOI of 0.01, and with LB broth as the control group. The mixture was incubated at 37°C for 15 min and then centrifuged at 8,000 rpm at 4°C for 2 min. The titer of free phage particles in the supernatant was measured through the double-layer plate method. Phage adsorption rate = [(initial phage titer − the supernatant phage titer)/initial phage titer] × 100%.

## Bacterial growth curve assay

Host strain and phage-resistant strains (200 µL; $10^6$ CFU/mL) were added into a 96-well plate and incubated at 37°C. $OD_{600}$ values were detected every 1 h and continuously monitored for 12 h.

## Biofilm assays of bacteria

Host strain and phage-resistant strains (200 µL; $10^8$ CFU/mL) were added to a 96-well plate and incubated at 37°C for 48 h. Wells were washed three times with PBS to remove planktonic bacteria. Then, 0.1% crystal violet was added to stain for 20 min. Wells were washed again with PBS and were placed in a ventilated and cool place to air dry. Ninety-five percent ethanol was added and decolorized for 20 min to completely dissolve the crystal violet. Absorbance values of each well were measured at 595 nm.

## Gene complementation

To confirm the role of *itrA3* and *gtr50* in phage susceptibility, we performed genetic complementation in the corresponding phage-resistant strains as described previously (27, 45). Both genes were PCR-amplified from the chromosomal DNA of *A. baumannii* T2784, digested with *EcoR*I and *Sal*I, and cloned into the vector pBAD18Kan-Ori. The resulting recombinant plasmids were electroporated (EC1, 1.0 kV) into the phage-resistant mutants. Following a 3-h recovery at 37°C, transformants were selected on LB agar supplemented with kanamycin (50 µg/mL). Individual colonies were grown to the exponential phase in LB broth, and gene expression was induced with 0.2% (wt/vol) arabinose. Phage adsorption assays were then conducted as described previously. The strains, plasmids, and primers used in this study are listed in Table S4.

## *In vitro* lysis assay

Bacteria and phages were added to the 96-well plate at an MOI of 100, 10, 1, 0.1, and 0.01 and then incubated at 37°C. $OD_{600}$ value was measured every 1 h to monitor the inhibitory effects of phages on bacterial growth.

## *In vivo* therapy assay

*Galleria mellonella* (*G. mellonella*) larvae were employed as an infection model to evaluate phage therapeutic efficacy. Prior to conducting phage therapy, *G. mellonella* infection models were established with different concentrations to determine the appropriate inoculation amount. A volume of 5 µL of bacteria at concentrations of $5 \times 10^8$, $2.5 \times 10^8$, $1 \times 10^8$, $5 \times 10^7$, $2.5 \times 10^7$, and $1 \times 10^7$ CFU/mL or PBS was injected into *G. mellonella* larvae. The amount of bacteria was determined using the standard plate count method. Then

the larvae were incubated at 37°C in dark conditions and were monitored every 12 h for 72 h. An optimal infection dose was selected for subsequent phage therapy experiments.

Following the determination of the optimal infection dose, the therapeutic effects of phages were evaluated. After 1 h of bacterial infection, phage was administered at an MOI of 100, 1, and 0.01. The larvae treated with phages were incubated at 37°C under dark conditions, with survival monitored and recorded every 12 h for 72 h as described above.

## Virulence assessment of bacteria

The virulence of bacteria was evaluated using the *G. mellonella* infection model. Bacteria were washed three times with PBS and serially diluted to appropriate concentrations. The wild-type host strain and phage-resistant strains (5 µL) were injected into larvae ($n =$ 10) and then incubated at 37°C in dark conditions. The mortality of *G. mellonella* in each group was recorded every 12 h for 72 h.

## Genomic DNA extraction and sequencing analysis

The λ phage genomic DNA extraction kit (Beijing Leadene Biotechnology Co., Ltd., China) and the bacterial genomic DNA extraction kit (Solarbio, China) were used to extract phage DNA and bacterial genomic DNA, respectively. All experimental procedures were carried out in compliance with the manufacturer's protocol. Phage and bacterial genomic DNA were subjected to next-generation sequencing at Novogene Bioinformatics Technology Co., Ltd. The clean data were assembled using SPAdes software (v3.13.0) for sequence assembly (46).

For phage genomes, BLASTn was used to compare with other phages in GenBank. MAFFT software (v7.515) was used to perform homology alignment of structural domain amino acid sequences. FastTree software (v2.1.11) was used to visualize the evolutionary tree (47). VRIDIC (https://rhea.icbm.uni-oldenburg.de/viridic/) was used to analyze the ANI between phages (48). ORFs were predicted and annotated using the online tool RAST (https://rast.nmpdr.org/rast.cgi) (49) and determined with NCBI's protein BLAST (BLASTp) and online tool InterPro (https://www.ebi.ac.uk/interpro/) (50). CG view (https://proksee.ca/) was used to draw a complete genome annotation circle diagram (51). Antimicrobial resistance genes and virulence factors were predicted using ResFinder (https://genepi.food.dtu.dk/resfinder) and VirulenceFinder (https://cge.food.dtu.dk/services/VirulenceFinder/).

For bacterial genome, PubMLST (52) (https://pubmlst.org/) was used to predict the multilocus sequence type of *A. baumannii*. Bautype (53) (http://bautype.net/Acinetobacter_baumannii/tools/) was used to identify capsule polysaccharide type of *A. baumannii*. Breseq (v0.39.0) was used to analyze the mutations present in phage-resistant isolates.

## Statistical analysis

Statistical analyses were performed using GraphPad Prism software (v10.1.2). Data were evaluated by one-way ANOVA, followed by Dunnett's *post hoc* test for multiple comparisons, with triplicate measurements presented as mean ± standard deviation (SD). A *P* value below 0.05 was considered statistically significant.

## ACKNOWLEDGMENTS

This work was supported by the Beijing Natural Science Foundation (F252052), the Peking University Third Hospital Fund for Interdisciplinary Research (BYSYJC2023005 and BYSYJC2024020), and the State Key Laboratory of Vascular Homeostasis and Remodeling Open Research Fund (2025-VHR-O-SY-21).

Z.Z. conducted the majority of the experimental work, including phage isolation and characterization, bioinformatic analyses, and drafted the initial manuscript. H.F. wrote and revised the manuscript. Z.H. contributed to the manuscript refinement. J.Z., N.S.,

Y.T., and M.L. designed and supervised the research and participated in manuscript finalization. All authors contributed to the article and approved the final version for publication.

## AUTHOR AFFILIATIONS

[1]Department of Laboratory Medicine, Peking University Third Hospital, Beijing, China
[2]Department of Pulmonary and Critical Care Medicine, Peking University Third Hospital, Beijing, China
[3]State Key Laboratory of Green Biomanufacturing, College of Life Science and Technology, Beijing University of Chemical Technology, Beijing, China

## AUTHOR ORCIDs

Zihe Zhou http://orcid.org/0009-0002-9502-0858
Mengzhe Li http://orcid.org/0000-0002-4390-8989
Yigang Tong http://orcid.org/0000-0002-8503-8045
Ning Shen http://orcid.org/0009-0007-8409-9835
Jiajia Zheng http://orcid.org/0000-0003-3939-9875

## FUNDING

| Funder | Grant(s) | Author(s) |
| --- | --- | --- |
| Natural Science Foundation of Beijing Municipality | F252052 | Jiajia Zheng |
| peking university third hospital fund for interdisciplinary research | BYSYJC2023005, BYSYJC2024020 | Jiajia Zheng |
| state key laboratory of vascular homeostasis and remodeling open research fund | 2025-VHR-O-SY-21 | Jiajia Zheng |

## AUTHOR CONTRIBUTIONS

Zihe Zhou, Data curation, Formal analysis, Validation, Visualization, Writing – original draft, Writing – review and editing | Hanyu Fu, Data curation, Writing – original draft, Writing – review and editing | Zhongyu Han, Conceptualization, Visualization, Writing – review and editing | Mengzhe Li, Conceptualization, Supervision, Visualization, Writing – review and editing | Yigang Tong, Conceptualization, Supervision, Visualization, Writing – review and editing | Ning Shen, Conceptualization, Supervision, Visualization, Writing – review and editing | Jiajia Zheng, Conceptualization, Funding acquisition, Supervision, Visualization, Writing – review and editing

## DATA AVAILABILITY

The complete genome sequence of phage pT2784 has been submitted to GenBank under accession number PV550753. The sequencing data of *Acinetobacter baumannii* T2784 and the three phage-resistant strains have been deposited in the NCBI database under BioProject accession number PRJNA1363103.

## ADDITIONAL FILES

The following material is available online.

### Supplemental Material

**Supplemental tables (Spectrum03301-25-s0001.xlsx).** Tables S1 to S4.

## Open Peer Review

**PEER REVIEW HISTORY (review-history.pdf).** An accounting of the reviewer comments and feedback.

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
