## [Reviewer comments · Microbiology Spectrum]

Microbiology Spectrum

The therapeutic potential of phage pT2784 against ST40-KL47 type *Acinetobacter baumannii* and bacterial fitness trade-offs

Mengzhe Li, Yigang tong, Ning Shen, jiajia Zheng, Zihe Zhou, Hanyu Fu, and Zhongyu Han

Corresponding Author(s): Mengzhe Li, Beijing University of Chemical Technology

Review Timeline:

Submission Date:	October 14, 2025
Editorial Decision:	November 4, 2025
Revision Received:	January 5, 2026
Accepted:	January 31, 2026

Editor: Fei Chen

Reviewer(s): Disclosure of reviewer identity is with reference to reviewer comments included in decision letter(s). The following individuals involved in review of your submission have agreed to reveal their identity: Wu nannan (Reviewer #1)

Transaction Report:

DOI: <https://doi.org/10.1128/spectrum.03301-25>

Re: Spectrum03301-25 (**The therapeutic potential of phage pT2784 against ST40-KL47 type *Acinetobacter baumannii* and bacterial fitness trade-offs**)

Dear Dr. Mengzhe Li:

Thank you for the privilege of reviewing your work. Below you will find my comments, instructions from the Spectrum editorial office, and the reviewer comments.

Revision Guidelines

Sincerely,
Fei Chen
Editor
Microbiology Spectrum

Reviewer #1 (Comments for the Author):

1. Introduction section: The second and third paragraphs of the Introduction were suggested to be replaced by a brief paragraph to the epidemiology of *Acinetobacter baumannii* genotypes (to suggest why those CPS types are selected in this paper), with a focus on KL47 *A. baumannii*, its epidemiology, drug resistance, treatment difficulties, and whether there are phage reports of this bacterium. Another paragraph introduces the phage of *A. baumannii* and their common mechanism of anti-phage resistance.
2. Results section:
 - 1) Reduce the overlapping parts from the Materials and methods section. Similarly, it would be appropriate to transfer some of

the background in the Discussion section to Introduction section.

- 2) Add the brief information about the host bacterium, such as its antibiotic susceptibility results.
3. table 1 should add a note: Use MLST-Pasteur typing.
4. line 144, it is suggested to specify the taxonomic position of phage pT2784.
5. line 214-216, uses the amino acid position within the mutant gene.
6. line 348, For culturation, bacteria were incubated...
7. line 373, pH
8. It is recommended to upload the sequences of isolated hosts and phage-resistant strains to NMDC or NCBI.

Reviewer #2 (Comments for the Author):

Recommendation: Major Revision

The manuscript by Zhou et al. describes the isolation and characterization of a novel bacteriophage, pT2784, active against a specific lineage of *Acinetobacter baumannii*. The study is well-structured and addresses a topic of significant interest in the field of alternative antimicrobials. The finding of a fitness trade-off, where phage resistance leads to attenuated virulence, is a valuable contribution. However, several key issues need to be addressed to strengthen the mechanistic conclusions and the perceived impact of the work. The points below should be considered for a revision.

Major Concerns:

1. The host range analysis is limited, testing only 20 *A. baumannii* strains across 8 ST-KL types. The small sample size and the lack of inclusion of a broader panel of multidrug-resistant (MDR) or extensively drug-resistant (XDR) clinical isolates limit the assessment of the phage's practical therapeutic potential. Furthermore, the absence of cross-infectivity tests against other common nosocomial genera (e.g., *Pseudomonas*, *Klebsiella*) leaves the possibility of non-specific lysis unaddressed. If possible, the host range should be expanded. If strain availability is a limiting factor, this must be explicitly acknowledged and discussed as a limitation in the "Discussion" section, with a statement on the future work required to define the therapeutic spectrum.
2. While missense mutations in the capsule synthesis genes *itrA3* and *gtr50* were identified in phage-resistant strains, the study relies on correlative evidence. Functional validation experiments (e.g., gene complementation, knockout) are necessary to directly prove that these specific mutations are the cause of the resistant phenotype. If complementation is not feasible, a more thorough discussion acknowledging this limitation is required. Additionally, the "Discussion" should be expanded to consider and rule out (based on genomic data or other evidence) other potential resistance mechanisms, such as CRISPR-Cas systems or restriction-modification systems.
3. The genomic annotation is incomplete, with 49 ORFs remaining as "hypothetical proteins." The study would benefit from a more in-depth bioinformatic analysis using tools for structural prediction and conserved domain analysis (e.g., HHPred, InterProScan) to assign putative functions. Please supplement the genomic analysis with deeper functional predictions for the hypothetical proteins and include the results in the manuscript or supplementary materials. Furthermore, to robustly support the claim of a new genus, additional phylogenetic analysis, such as a proteomic tree based on key conserved proteins (e.g., terminase large subunit), should be provided.

Minor Points and Editorial Corrections:

The following specific language and formatting issues should be corrected throughout the manuscript:

1. (P4, L77-78): Subject-verb agreement error. Correct "has discouraged" to "have discouraged".
2. (P2, L22): Article usage. Use "the" World Health Organization priority pathogen.
3. (P5, L106): Phrasing. Suggest revising to "a novel phage, designated as pT2784, was isolated".
4. (P10, L198-200): Sentence fragment. Revise the sentence "Considering that..., resulting in..." to form a complete sentence, e.g., "As this mechanism results in..., we first...".
5. (P13, L223-224): Redundancy. Correct "early-stage pathway of capsule synthesis pathway" to "early stage of capsule synthesis".
6. (P7, L134): Preposition usage. Correct "representing it a potential candidate" to "representing it as a potential candidate".
7. Formatting: Ensure a space is placed between all numerical values and their units (e.g., "5 μ L", "1 h", "10 min").
8. References: Ensure the reference list is formatted uniformly according to the journal's style guide.

Dear Editors and Reviewers:

Thank you very much for your valuable comments and suggestions regarding our manuscript entitled "The therapeutic potential of phage pT2784 against ST40-KL47 type *Acinetobacter baumannii* and bacterial fitness trade-offs" (Manuscript ID: Spectrum03301-25). These comments are immensely helpful in improving the quality of our work and have provided us with significant insights. We have carefully considered all suggestions and have revised the manuscript accordingly. We provide a point-by-point response to each comment as follows. All changes in the revised manuscript have been highlighted **in yellow** for your convenience.

Reviewer #1 (Comments for the Author):

*1. Introduction section: The second and third paragraphs of the Introduction were suggested to be replaced by a brief paragraph to the epidemiology of *Acinetobacter baumannii* genotypes (to suggest why those CPS types are selected in this paper), with a focus on KL47 *A. baumannii*, its epidemiology, drug resistance, treatment difficulties, and whether there are phage reports of this bacterium. Another paragraph introduces the phage of *A. baumannii* and their common mechanism of anti-phage resistance.*

Response: Thanks so much for your valuable suggestion. As suggested, we have provided a comprehensive overview of the molecular epidemiology of *Acinetobacter baumannii*, highlighting that the ST40 strain used in our study represents a significant emerging clone, in the revised second paragraph of Introduction. Additionally, we emphasized that the KL47 capsular type is widely distributed not only in the predominant ST2 lineage but also in other clinically prevalent lineages like ST40 and ST164, which pose a substantial threat to healthcare settings. The limited availability of phages targeting the KL47 type was also explained here. And in the third paragraph, we have added the description about current applications of *A. baumannii* phages and common mechanisms of anti-phage resistance. The details are as follows:

“The remarkable genetic plasticity of *A. baumannii* has resulted in extensive strain diversity among clinical isolates (7-11). Epidemiological surveillance in China has identified ST2_Pasteur through multi-locus sequence typing (MLST) as the predominant clone responsible for hospital-acquired *A. baumannii* infections. ST40_Pasteur and ST164_Pasteur has emerged as an increasingly prevalent clone, with its clinical isolation rate continuous increase in the past three years (7, 12-14). The gene encoding capsule polysaccharides (CPS) on the K locus is strain specific, allowing for a huge diversity in the composition of *A. baumannii* capsules. Current reports have documented at least 120 distinct K locus variants and approximately 40 structurally unique polysaccharides (8, 15). The KL47 capsular type is widely distributed not only in the predominant ST2 lineage but also in other clinically prevalent lineages, including ST40 and ST164. Critically, KL47 *A. baumannii* isolates within these lineages frequently demonstrate a multidrug-resistant phenotype (16, 17), posing a significant threat to healthcare systems. *A. baumannii* phages primarily utilized the CPS as their receptors (18, 19). Although over ten CPS-targeting phages have been characterized, such as phage Highwayman targeting ST2-KL3 strains, phages Margaret and Fishpie infecting ST2-KL2 strains, and phage Tama, which could recognize ST2-KL9 capsule type (18), phage resources specifically targeting the KL47 stains remain remarkably scarce.

Accumulating evidence demonstrates that phages, whether administered as monotherapy or in combination with antibiotics, can effectively combat *A. baumannii* infections (20-24). Despite its considerable therapeutic promise, phage therapy remains constrained by substantial challenges that impede its broad clinical adoption. The expeditious development of phage resistance poses a central obstacle, stemming from either the structural alteration of surface receptors or the deployment of intracellular defense systems including restriction modification (R-M) and abortive infection (Abi) systems (25, 26). Mutations that alter or eliminate phage receptors constitute the primary mechanism of phage resistance in *A. baumannii* (27-29).” (Line 66-93)

2. Results section:

1) Reduce the overlapping parts from the Materials and methods section. Similarly, it would be appropriate to transfer some of the background in the Discussion section to Introduction section.

Response: Thanks for your valuable suggestion. We have carefully reviewed the Results section and revised the text to reduce overlapping methodological descriptions. The changes were mainly focused on the latter half of the Results. The revision to **the Results section** (Line 168-226) is as follows:

“In vitro and in vivo efficacy of phages against *A. baumannii*

Phage pT2784 exhibited strong lytic activity against host bacteria in vitro (Figure 4A), completely suppressing bacterial growth at first across a range of MOIs. Notably, after 6 h of incubation, an increase in optical density at 600 nm (OD600) was observed, despite much more slowly than the bacteria-only group. This phenomenon suggested phage-resistant bacterial variants could emerge after long-term co-incubation.

In a *G. mellonella* infection model, inoculation with 5×10^7 CFU/mL *A. baumannii* resulted in progressive larval mortality, with a final survival rate of only 10% at 72 h post-infection (Figure 4B). This dose provided a suitable observation window and was therefore selected for subsequent in vivo efficacy experiments.

Treatment with phage pT2784 significantly enhanced the survival of infected *G. mellonella* larvae in a dose-dependent manner, with the greatest protection observed at a high MOI of 100 (Figure 4C). Notably, larvae receiving phage alone exhibited 100% survival, confirming the in vivo safety of pT2784 and underscoring its potential as an alternative antimicrobial agent.

Impairment of early-stage pathway of capsule synthesis confers resistance to phages

To investigate the resistance mechanisms, three phage-resistant isolates, designated pT2784-R1, R2, and R3, were obtained from the 24-h co-culture. All three isolates exhibited resistance, as shown by the absence of plaques (Figure 5A) and their growth on soft agar containing phage pT2784 (Figure 5B).

As receptor modification or loss constitutes the primary molecular mechanism resulting in impaired phage adsorption (27, 33), we first quantitatively compared the adsorption efficiency of phage pT2784 between wild-type and phage-resistant strains. In contrast to the near-complete

adsorption observed in the wild-type host, the adsorption efficiencies in the phage-resistant isolates pT2784-R1, pT2784-R2, and pT2784-R3 dropped sharply to 19.15%, 20.84%, and 24.01%, respectively (Figure 5C, D, E). This marked reduction is consistent with receptor loss or structural modification as the underlying resistance mechanism.

Comparative genomic analysis identified single-nucleotide mutations in the *itrA3* gene of pT2784-R1 (Leu to Pro at position 151) and pT2784-R3 (Gly to Arg at position 79), as well as in the *gtr50* gene of pT2784-R2 (Val to Glu at position 225) (Figure 5F). The *itrA3* and *gtr50* genes were located within the K locus. Specifically, *itrA3* encoded the initial glycosyltransferase responsible for oligosaccharide unit initiation (34), while *gtr50* encoded a glycosyltransferase (35). These mutations likely affected the function of the encoded enzymes, impairing the early stage of capsule synthesis and conferring phage resistance. This genetic evidence strongly supports the capsular polysaccharide as the primary receptor for phage pT2784 adsorption and subsequent infection.

Genetic complementation confirmed that *itrA3* and *gtr50* mutations are responsible for phage resistance. pT2784-R1 and pT2784-R3 with the introduction of the *itrA3* gene and pT2784-R2 complemented with *gtr50* restored phage adsorption efficiencies ($P > 0.05$, Figure 5C, D, E).

Attenuation of biofilm formation and virulence in phage-resistant isolates

Given the critical role of capsular polysaccharide as a major surface structure in *A. baumannii*, we next assessed whether its structural alterations imposed fitness trade-offs on phage-resistant strains (36-38). The phage-resistant isolates and the wild-type strain formed similar circular colonies with gray white, smooth, and neat edges on blood agar, with no apparent morphological differences (Figure 6A). Growth rates of phage-resistant isolates also showed no significant difference compared to that of the wild-type strain (Figure 6B).

Scanning electron microscopy (SEM) revealed that all phage-resistant isolates exhibited increased surface rugosity compared to the wild-type strain (Figure 6C). Furthermore, biofilm formation by the resistant isolates was significantly reduced relative to the wild-type after 48 hours of cultivation (Figure 6D).

In a *G. mellonella* infection model (Figure 6E), larvae inoculated with the wild-type strain at a concentration of 1×10^8 CFU/mL exhibited 100% mortality within 12 hours. In contrast, infection with phage-resistant strains resulted in less than 20% mortality at 12 hours, with final mortality not exceeding 50% after 72 hours, indicating a significant attenuation of virulence in the resistant variants.”

Additionally, we have relocated the first paragraph of **the Discussion section** to the Introduction. So, the revised Discussion now begins with "In this study, we isolated phage pT2784...". (Line 228)

2) Add the brief information about the host bacterium, such as its antibiotic susceptibility results.

Response: Thanks for your reminder. As suggested, we have added a brief introduction of

antibiotic susceptibility results of the host bacterium as follows: “The host strain T2784 was isolated from a patient's respiratory sample and was relatively susceptible to commonly used antibiotics. Other strains of the same ST40-KL47 lineage have also been recovered from the patient’s bed unit environment.” (Line 115-118) Detailed antibiotic susceptibility information is in Supplementary Table S1.

We have also added relevant descriptions in **the Discussion section**. The details were as follows:

“Although the host strain T2784 obtained in this study is relatively susceptible to commonly used antibiotics, it is noteworthy that both the ST40 and KL47 lineages have been widely associated with multidrug-resistant phenotypes. The convergence of these two high-risk lineages in a single strain suggests a potential risk for the evolution of resistance (Supplementary Table S3). Consequently, isolating and characterizing phages targeting such strains is necessary for future therapeutic and preventive strategies. Notably, phages specific to the ST40-KL47 type of *A. baumannii* have rarely been reported, thus expanding the phage resource library against this lineage is of considerable importance.” (Line 282-290)

3. table 1 should add a note: Use MLST-Pasteur typing.

Response: Thanks for your kind reminder. We have added a note to Table 1 as follows: "NOTE: MLST-Pasteur typing was used. MDR, multidrug-resistant; XDR, extensively drug-resistant." (Line 121-122)

4. line 144, it is suggested to specify the taxonomic position of phage pT2784.

Response: Thanks for your suggestion. We have revised the sentence to explicitly state: "supporting its placement within a novel genus under *Caudoviricetes*; *Caudoviricetes incertae sedis*." (Line 142-143)

5. line 214-216, uses the amino acid position within the mutant gene.

Response: Thanks for your suggestion. The mutations are now described with their amino acid positions as follows: “single-nucleotide mutations in the *itrA3* gene of pT2784-R1 (Leu to Pro at position 151) and pT2784-R3 (Gly to Arg at position 79), as well as in the *gtr50* gene of pT2784-R2 (Val to Glu at position 225).” (Line 197-199)

6. line 348, For culturation, bacteria were incubated...

Response: Thanks for your suggestion. We have revised the sentence as follows: “For culturation, bacteria were incubated in Luria-Bertani (LB) broth overnight at 37°C with shaking at 200 rpm.” (Line 307-308)

7. line 373, pH

Response: Thanks for your reminder. We have revised “PH and temperature stability” to “pH and temperature stability”. (Line 334)

8. It is recommended to upload the sequences of isolated hosts and phage-resistant strains to NMDC or NCBI.

Response: Thanks for your suggestion. We have deposited the sequences of the host and three phage-resistant strains in the NCBI under the following accession numbers, SAMN49984799, SAMN49984800, SAMN49984801, and SAMN49984802. Meanwhile, we also mentioned it in the Data Availability Statement as follows:

“The sequencing data of *A. baumannii* T2784 and three phage-resistant strains were submitted to the NCBI database under the accession numbers SAMN49984799, SAMN49984800, SAMN49984801, SAMN49984802.” (Line 457-460)

Reviewer #2 (Comments for the Author):

Recommendation: Major Revision

The manuscript by Zhou et al. describes the isolation and characterization of a novel bacteriophage, pT2784, active against a specific lineage of Acinetobacter baumannii. The study is well-structured and addresses a topic of significant interest in the field of alternative antimicrobials. The finding of a fitness trade-off, where phage resistance leads to attenuated virulence, is a valuable contribution. However, several key issues need to be addressed to strengthen the mechanistic conclusions and the perceived impact of the work. The points below should be considered for a revision.

Response: Thank you very much for your insightful comments on our manuscript. We have carefully revised the manuscript in accordance with your suggestions to enhance its clarity and scientific quality. Our point-by-point responses are as follows. We hope that these revisions have fully addressed your concerns and questions.

Major Concerns:

1. The host range analysis is limited, testing only 20 A. baumannii strains across 8 ST-KL types. The small sample size and the lack of inclusion of a broader panel of multidrug-resistant (MDR) or extensively drug-resistant (XDR) clinical isolates limit the assessment of the phage's practical therapeutic potential. Furthermore, the absence of cross-infectivity tests against other common nosocomial genera (e.g., Pseudomonas, Klebsiella) leaves the possibility of non-specific lysis unaddressed. If possible, the host range should be expanded. If strain availability is a limiting factor, this must be explicitly acknowledged and discussed as a limitation in the "Discussion" section, with a statement on the future work required to define the therapeutic spectrum.

Response: Thanks so much for your constructive suggestion. As suggested, we have significantly expanded the host range analysis. The testing panel now includes 36 *A. baumannii* clinical isolates spanning 19 distinct ST-KL types, the majority of which are MDR or XDR. Furthermore, we have explicitly tested for cross-genus infectivity by including several *Pseudomonas aeruginosa* and *Klebsiella pneumoniae* strains. The antibiotic susceptibility tests for all strains were provided in

Supplementary Table S1. The revised details were as follows:

“The host range analysis of phage pT2784 was performed against a diverse panel of 36 *A. baumannii* clinical isolates spanning 19 distinct ST-KL types, which included both multidrug-resistant (MDR)/extensively drug-resistant (XDR) and susceptible strains. Notably, the phage exhibited strict specificity for ST40-KL47 type *A. baumannii* strains, while it showed no infectivity towards the other genotypes and other common nosocomial pathogens such as *Klebsiella pneumoniae* (*K. pneumoniae*) and *Pseudomonas aeruginosa* (*P. aeruginosa*), as detailed in Table 1. The host strain T2784 was isolated from a patient's respiratory sample and was relatively susceptible to commonly used antibiotics. Other strains of the same ST40-KL47 lineage have also been recovered from the patient's bed unit environment. Antibiotic susceptibility profiles of all strains used in host range test are provided in Supplementary Table S1.

Table 1 Host range of pT2784

Species	Strain	ST type	KL type	MDR/XDR Status	Phage sensitivity
A. baumannii	T12965	2	2	XDR	-
A. baumannii	T12967	2	2	XDR	-
A. baumannii	T13520	2	2	XDR	-
A. baumannii	T3048	2	3	XDR	-
A. baumannii	T3094	2	3	XDR	-
A. baumannii	T3307	2	3	XDR	-
A. baumannii	T1319	2	7	XDR	-
A. baumannii	T1658	2	7	XDR	-
A. baumannii	T2010	2	7	XDR	-
A. baumannii	T12430	2	9	XDR	-
A. baumannii	T12457	2	9	XDR	-
A. baumannii	T13166	2	9	XDR	-
A. baumannii	T2297	40	45	XDR	-
A. baumannii	T2784	40	47	-	+
A. baumannii	T3153	40	47	-	+
A. baumannii	T3311	113	139	MDR	-
A. baumannii	T3840	113	139	MDR	-
A. baumannii	T4258	113	139	MDR	-
A. baumannii	T10332	571	10	XDR	-
A. baumannii	T10408	571	10	XDR	-
A. baumannii	T11233	2	104	XDR	-
A. baumannii	T1058	2	104	XDR	-
A. baumannii	T2977	2	104	XDR	-
A. baumannii	T10252	2	101	XDR	-
A. baumannii	T10331	2	101	XDR	-
A. baumannii	T11889	2	101	XDR	-
A. baumannii	T13697	2	160	XDR	-
A. baumannii	T1005	2	160	MDR	-
A. baumannii	T11280	221	14	MDR	-

A. baumannii	T10858	284	14	-	-
A. baumannii	T11572	396	38	MDR	-
A. baumannii	T10236	464	210	XDR	-
A. baumannii	T10780	119	100	MDR	-
A. baumannii	T11867	57	230	-	-
A. baumannii	T10223	354	125	-	-
A. baumannii	F7320	23	108	-	-
K. pneumoniae	217370	11	47	XDR	-
K. pneumoniae	T5003	11	64	XDR	-
P. aeruginosa	T3475	162	-	MDR	-
P. aeruginosa	T3553	1196	-	-	-

NOTE: MLST-Pasteur typing was used. MDR, multidrug-resistant; XDR, extensively drug-resistant.” (Line 109-120)

2. While missense mutations in the capsule synthesis genes *itrA3* and *gtr50* were identified in phage-resistant strains, the study relies on correlative evidence. Functional validation experiments (e.g., gene complementation, knockout) are necessary to directly prove that these specific mutations are the cause of the resistant phenotype. If complementation is not feasible, a more thorough discussion acknowledging this limitation is required. Additionally, the "Discussion" should be expanded to consider and rule out (based on genomic data or other evidence) other potential resistance mechanisms, such as CRISPR-Cas systems or restriction-modification systems.

Response: Thanks so much for your constructive suggestion. We performed gene complementation of the *itrA3* and *gtr50* genes. Restoring the wild-type *itrA3* gene in resistant strains pT2784-R1 and pT2784-R3 recovered phage adsorption efficiency, and complementing *gtr50* in strain pT2784-R2 fully restored adsorption. These results demonstrate that these specific mutations are directly responsible for the phage resistance phenotype. Consequently, we have added relevant descriptions in the Results and Discussion sections. We also supplemented the details of gene complementation in the Materials and methods section. As the mechanism has been definitively established through these experiments, we have not extended the discussion to other potential mechanisms in the revised manuscript. The revised details are as follows:

Results

“Genetic complementation confirmed that *itrA3* and *gtr50* mutations are responsible for phage resistance. pT2784-R1 and pT2784-R3 with the introduction of the *itrA3* gene and pT2784-R2 complemented with *gtr50* restored phage adsorption efficiencies ($P > 0.05$, Figure 5C, D, E).” (Line 206-209)

Discussion

“Disruption of these genes likely impairs capsule assembly and alters surface receptor structures, thereby reducing phage adsorption. This effect was nearly fully reversed upon genetic complementation (Figure 5C, D, E).” (Line 258-260)

Materials and methods

“Gene complementation

To confirm the role of *itrA3* and *gtr50* in phage susceptibility, we performed genetic complementation in the corresponding phage-resistant strains as described previously (27, 45). Both genes were PCR-amplified from the chromosomal DNA of *A. baumannii* T2784, digested with *EcoRI* and *Sall*, and cloned into the vector pBAD18Kan-Ori. The resulting recombinant plasmids were electroporated (EC1, 1.0 kV) into the phage-resistant mutants. Following a 3-h recovery at 37°C, transformants were selected on LB agar supplemented with kanamycin (50 µg/mL). Individual colonies were grown to exponential phase in LB broth, and gene expression was induced with 0.2% (w/v) arabinose. Phage adsorption assays were then conducted as described previously. The strains, plasmids, and primers used in this study are listed in Supplementary Table S4.” (Line 383-394)

3. The genomic annotation is incomplete, with 49 ORFs remaining as "hypothetical proteins." The study would benefit from a more in-depth bioinformatic analysis using tools for structural prediction and conserved domain analysis (e.g., HHPred, InterProScan) to assign putative functions. Please supplement the genomic analysis with deeper functional predictions for the hypothetical proteins and include the results in the manuscript or supplementary materials. Furthermore, to robustly support the claim of a new genus, additional phylogenetic analysis, such as a proteomic tree based on key conserved proteins (e.g., terminase large subunit), should be provided.

Response: Thank you so much for suggestion. We have performed an in-depth analysis of the 49 hypothetical proteins using InterProScan. Inspiringly, we have assigned putative functions to 12 ORFs (highlighted in yellow in Supplementary Table S2). By applying a lower BLASTp identity threshold, putative functions were also predicted for an additional 4 ORFs (highlighted in orange in Supplementary Table S2). Furthermore, as recommended, we constructed a phylogenetic tree based on the terminase large subunit (Figure 2D). Combined with the ANI value below 70%, this robustly supports the classification of phage pT2784 as a novel genus. We have also supplemented the relevant details in **the Materials and Methods** section. The revised details are as follows:

Materials and Methods

“determined with NCBI's protein BLAST (BLASTp) and online tool InterPro (<https://www.ebi.ac.uk/interpro/>)” (Line 433-434)

Results

“Phage pT2784 had a DNA genome with a length of 44335 bp and a GC content of 37.82% (Figure 2A). Genomic annotation predicted 76 putative open reading frames (ORFs), among which 43 encoded functional proteins (Table 2). These ORFs were categorized into five essential processes: phage structural composition, DNA replication and repair, transcription and translation, phage packaging, and host lysis. These included 21 structural proteins (ORF 17, ORF27-29, ORF31-32, ORF34, ORF36-40, ORF43-46, ORF48-52), 9 proteins associated with transcription

and translation (ORF4, ORF6, ORF11, ORF33, ORF56, ORF61, ORF64-65, ORF74), 3 packaging proteins (ORF14-16), 6 proteins related to DNA replication and repair (ORF2, ORF35, ORF60, ORF68, ORF70-71), and 4 host lysis protein (ORF41, ORF42, ORF53-54). The remaining ORFs encoded hypothetical proteins with unknown functions (Supplementary Table S2). No lysogenic related genes found in the genome of pT2784, representing it as a potential candidate with clinical application values.” (Line 124-136)

“While phylogenetic analysis based on the terminase large subunit (Figure 2D), which included the 13 most similar sequences, placed pT2784 within the same clade as *Obolenskivirus* phage Abp95 (MZ618622.1). Despite this conserved functional gene association, the low whole-genome ANI supported the classification of pT2784 as a novel genus.

Table 2 Putative functional ORFs in phage pT2784 genome

ORF	Strand	Start	Stop	Nucleotide length (bp)	Amino acid length (aa)	Putative function
ORF2	+	278	433	156	52	nucleotide kinase
ORF4	+	944	1165	222	74	RusA family crossover junction endodeoxyribonuclease
ORF6	+	1608	1823	216	72	DUF551 domain-containing protein
ORF11	+	3462	3662	201	67	HTH DNA binding protein
ORF14	+	4810	5241	432	144	terminase small subunit
ORF15	+	5342	6523	1182	394	terminase large subunit
ORF16	+	6527	7954	1428	476	portal protein
ORF17	+	8188	8895	708	236	Head morphogenesis protein
ORF27	+	12902	14236	1335	445	head maturation

						protease
ORF28	+	14244	14723	480	160	minor head protein
ORF29	+	14733	15752	1020	340	major capsid protein
ORF31	+	16251	16622	372	124	head tail connector
						protein
ORF32	+	16638	16784	147	49	tail length tape
						measure protein
ORF33	+	16829	17221	393	131	RNA polymerase
ORF34	+	17308	17403	96	32	neck protein
ORF35	+	17477	18034	558	186	DNA polymerase III
						beta subunit
ORF36	+	18072	18575	504	168	head adaptor
ORF37	+	18602	19066	465	155	tail completion protein
ORF38	+	19149	20519	1371	457	tail sheath protein
ORF39	+	20532	20981	450	150	virion structural
						protein
ORF40	+	21027	21452	426	142	tail assembly
						chaperone
ORF41	+	21482	21694	213	71	putative
						tail-fiber/lysozyme
						protein
ORF42	+	21697	23727	2031	677	lysozyme-like protein
ORF43	+	23735	24331	597	199	tail tube initiator
ORF44	+	24333	24611	279	93	virion structural
						protein
ORF45	+	24798	25610	813	271	baseplate hub
ORF46	+	25765	26238	474	158	baseplate assembly
						protein

” (Line 145-149)

Minor Points and Editorial Corrections:

The following specific language and formatting issues should be corrected throughout the manuscript:

1. (P4, L77-78): *Subject-verb agreement error. Correct "has discouraged" to "have discouraged".*

Response: Thanks for your suggestion. We have thoroughly checked the relevant text. Therefore, the sentence containing this error was removed entirely in the revised manuscript, thus the issue has been resolved accordingly. Thank you again for your thoughtful and constructive reminder.

2. (P2, L22): *Article usage. Use "the" World Health Organization priority pathogen.*

Response: Thanks for your reminder. We have revised "World Health Organization priority pathogen" to "the World Health Organization priority pathogen". (Line 22)

3. (P5, L106): *Phrasing. Suggest revising to "a novel phage, designated as pT2784, was isolated".*

Response: Thanks for your reminder. We have revised as follows: "Using clinically isolated *A. baumannii* T2784 as the host, a novel phage, designated as pT2784, was isolated from hospital sewage samples." (Line 102-103)

4. (P10, L198-200): *Sentence fragment. Revise the sentence "Considering that..., resulting in..." to form a complete sentence, e.g., "As this mechanism results in..., we first..."*

Response: Thanks for your reminder. We have revised "Considering that..., resulting in significantly impaired phage adsorption. We first quantitatively compared the adsorption efficiency of phage pT2784 between wild-type and phage-resistant strains" to "As receptor modification or loss constitutes the primary molecular mechanism resulting in impaired phage adsorption (27, 33), we first quantitatively compared the adsorption efficiency of phage pT2784 between wild-type and phage-resistant strains." (Line 189-191)

5. (P13, L223-224): *Redundancy. Correct "early-stage pathway of capsule synthesis pathway" to "early stage of capsule synthesis".*

Response: Thanks for your reminder. We have revised "impairing the early-stage pathway of capsule synthesis pathway" to "impairing the early stage of capsule synthesis". (Line 203)

6. (P7, L134): *Preposition usage. Correct "representing it a potential candidate" to "representing it as a potential candidate".*

Response: Thanks for your reminder. We have revised "representing it a potential candidate" to "representing it as a potential candidate" (Line 136)

7. *Formatting: Ensure a space is placed between all numerical values and their units (e.g., "5 μ L", "1 h", "10 min").*

Response: Thanks for your reminder. We have checked the formatting throughout the manuscript and confirmed that a space is placed between all numerical values and their units. The text has been revised accordingly.

8. References: Ensure the reference list is formatted uniformly according to the journal's style guide.

Response: Thanks for your reminder. We have carefully revised the reference list throughout the manuscript to ensure strict adherence to the journal's formatting style. Each citation has been checked and updated accordingly.

Re: Spectrum03301-25R1 (**The therapeutic potential of phage pT2784 against ST40-KL47 type *Acinetobacter baumannii* and bacterial fitness trade-offs**)

Dear Dr. Mengzhe Li:

Your manuscript has been accepted, and I am forwarding it to the ASM production staff for publication. Your paper will first be checked to make sure all elements meet the technical requirements. ASM staff will contact you if anything needs to be revised before copyediting and production can begin. Otherwise, you will be notified when your proofs are ready to be viewed.

Sincerely,
Fei Chen
Editor
Microbiology Spectrum

Reviewer #2 (Comments for the Author):

Thank you for your thoughtful revision and supplementary improvements to this article.